# Exploring Conditional Shifts for Domain Generalization

## Abstract

Learning a domain-invariant representation has become one of the most popular approaches for domain adaptation/generalization. In this paper, we show that the invariant representation may not be sufficient to guarantee a good generalization, where the **labeling function shift** should be taken into consideration. Inspired by this, we first derive a new generalization upper bound on the empirical risk that explicitly considers the labeling function shift. We then propose **Domain-specific Risk Minimization (DRM)**, which can model the distribution shifts of different domains separately and select the most appropriate one for the target domain. Extensive experiments on four popular domain generalization datasets, CMNIST, PACS, VLCS, and DomainNet, demonstrate the effectiveness of the proposed DRM for domain generalization with the following advantages: 1) it significantly outperforms competitive baselines; 2) it enables either comparable or superior accuracies on all training domains comparing to vanilla empirical risk minimization (ERM); 3) it remains very simple and efficient during training, and 4) it is complementary to invariant learning approaches.

## 1 Introduction

Machine learning models usually suffer from degraded performance when the testing data comes from a different distribution than the training data. To overcome the brittleness of classical Empirical Risk Minimization (ERM), domain generalization (DG) approaches have recently been proposed Muandet et al. (2013); Li et al. (2018b), where models are trained on multiple large-scale source domains/datasets and can be deployed on unseen target domains directly without any data collection/annotation and/or model updating.

Deep learning has recently made a great success and most deep learning-based DG methods seek to learn an invariant representation which is not only robust to all training domains but also can generalize to unknown distributions that may have a shift from the training distribution Muandet et al. (2013); Arjovsky et al. (2019); Sagawa et al. (2019); Li et al. (2018b). Among them, multi-source domain adaptation and its disentangled variant have provided rigorous bounds for learning transferable representations with target domain data available (Zhao et al., 2018; Ruichu et al., 2019). However, without accessing the data on the target domain, feature alignment can be performed only among source domains, which inevitably raises a question: whether the representation invariant to the source domain shift is good enough to generalize on the unseen target domain?

To answer the above-mentioned question, we first construct a simple counterexample, where the invariant representation learned on source domains fails to generalize on the target domain (please see the details of this counterexample in the next section). This counterexample shows that without considering the labeling function shifts of different domains, even a perfect domain-invariant representation among source domains may also lead to very large errors on both source and target domains. To better understand the effect of labeling function shifts for DG, we further prove a new generalization upper bound that considers labeling function shifts between source and target domains. Specifically, an intuitive explanation of the new generalization upper bound is as follows: *since we cannot guarantee that all labeling functions are the same, we would rather model all labeling functions and choose the most appropriate one for a good generalization during testing.*

Therefore, in this paper, we introduce a shared encoder for all source domains with a group of domain-specific classifiers during training. Specifically, each domain-specific classifier is responsible for the labeling function

on a specific source domain. During testing, we further propose an entropy minimization strategy for classifier selection. That is, given each data sample from the target domain, we choose the classifier with the most confident prediction as to the final prediction. To this end, we devise a new method for domain generalization, **Domain-Specific Risk Minimization (DRM)**, which aims to reduce the negative impact of domain labeling function variations and can be easily incorporated into most deep representation learning algorithms. Our main contributions in this paper are as follows:

- **A new perspective.** Through a counterexample, we show the insufficiency of invariant representations and provide a new generalization bound that explicitly considers the conditional shift between source and target domains.

- **A new approach.** We propose a new Domain-specific Risk Minimization (DRM) method, which models all labeling functions in a domain-specific way during training, and then selects an appropriate labeling function for the target domain based on the entropy minimization strategy.

- **Extensive experiments.** We perform extensive experiments on popular DG benchmarks showing that DRM achieves very competitive performance and is orthogonal to other DG methods. Furthermore, case studies show that DRM not only beats IRM Arjovsky et al. (2019) on the average generalization performance but also reserves very strong recognition capability on source domains.

The rest of this paper is organized as follows. In Section 2, we analyze the failure cases of learning invariant representations for domain generalization, and provide a new generalization bound by explicitly taking into account label function shifts. After that, we present a domain-specific risk minimization (DRM) method in Section 3. We discuss the related work in Section 5 and the experimental results are shown in Section 4. Lastly, we conclude the paper in Section 6.

## 2 Preliminaries

Let $\mathcal{X}, \mathcal{Y}, \mathcal{Z}$ denote the input, output, and feature space, respectively. Let $X, Y, Z$ denote the random variables taking values from $\mathcal{X}, \mathcal{Y}, \mathcal{Z}$, respectively. Each domain corresponds to a joint distribution $P_i(X, Y)$ with a labeling function $f_i : \mathcal{X} \to [0, 1]$[1]. In the DG setting, we have access to a labeled training dataset which consists of several different but related training distributions (domains): $\mathcal{D} = \cup_{i=1}^{K} \mathcal{D}_i$, where $K$ is the number of domains. In this paper, we focus on a deterministic setting where the output $Y = f_i(X)$ is given by a deterministic labeling function, $f_i$, which varies from domain to domain. Let $g : \mathcal{X} \to \mathcal{Z}$ denote the encoder/feature transformation and $h : \mathcal{Z} \to \{0, 1\}$ denote the classifier/hypothesis. The error incurred by $h \circ g$ under domain $\mathcal{D}_i$ can be defined as $\epsilon_i(h \circ g) = \mathbb{E}_{X \sim \mathcal{D}_i}[|h \circ g(X) - f_i(X)|]$. Given $f_i$ and $h$ as binary classification functions, we have

$$\begin{aligned}
\epsilon_i(h \circ g) = \epsilon_i(h \circ g, f_i) &= \mathbb{E}_{X \sim \mathcal{D}_i}[|h \circ g(X) - f_i(X)|] \\
&= \Pr_{X \sim \mathcal{D}_i}(h \circ g(X) \neq f_i(X)).
\end{aligned} \tag{1}$$

During training, $h \circ g$ is trained using all image-label pairs from $\mathcal{D}$. During testing, we perform a retrieval task on the unseen target domain $\mathcal{D}_\mathcal{T}$ without additional model updating and we aim to minimize the error in $\mathcal{D}_\mathcal{T}$: $\min_{h \circ g} \epsilon_\mathcal{T}(h \circ g)$. This objective encodes the goal of learning a model that does not depend on spurious correlations (*e.g.,* domain-specific information): if a model makes decisions according to domain-specific information, it is natural to be brittle in an entirely distinct domain.

### 2.1 A Failure Case of Invariant Representation

In revealing flaws of learning invariant representations, we begin with a simple counterexample, where invariant representations fail to generalize. As shown in Fig. 1, given the following four domains: $\mathcal{D}_o \sim$

---

[1]Most theories and examples in this paper considers binary classification for easy understanding and can be easily extended to multi-class classification.

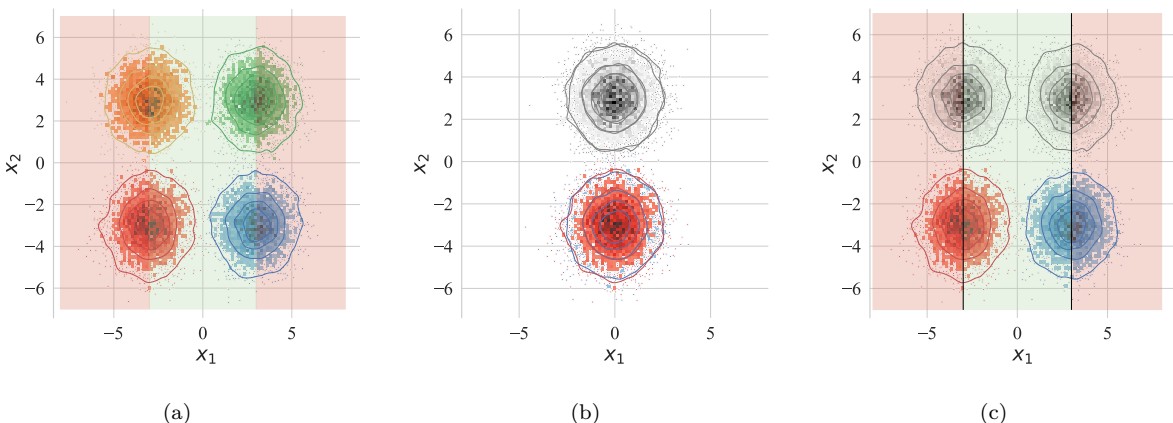

(a)                                                          (b)                                                          (c)

Figure 1: A failure case of invariant representations for domain generalization. (a) Given four domains in different colors, orange ($\mu_o = [-3.0, 3.0]$), green ($\mu_g = [3.0, 3.0]$), red ($\mu_r = [-3.0, -3.0]$) and blue ($\mu_b = [3.0, -3.0]$). (b) Invariant representations learnt from domain $\mathcal{D}_r$ and $\mathcal{D}_b$ by feature transformation $g(X) = \mathbb{I}_{x_1 < 0} \cdot (x_1 + 3) + \mathbb{I}_{x_1 > 0} \cdot (x_1 - 3)$. The grey color indicates the transformed target domains. (c) The classification boundary learnt by DRM.

$\mathcal{N}([-3, 3], I), \mathcal{D}_g \sim \mathcal{N}([3, 3], I), \mathcal{D}_r \sim \mathcal{N}([-3, -3], I), \mathcal{D}_b \sim \mathcal{N}([3, -3], I)$, where $X = (x_1, x_2)$ and

$$f_o(X) = \begin{cases} 0, & x_1 \leq -3 \\ 1, & x_1 > -3 \end{cases}, \quad f_r(X) = \begin{cases} 1, & x_1 \leq -3 \\ 0, & x_1 > -3 \end{cases},$$
$$f_g(X) = \begin{cases} 0, & x_1 \leq 3 \\ 1, & x_1 > 3 \end{cases}, \quad f_b(X) = \begin{cases} 1, & x_1 \leq 3 \\ 0, & x_1 > 3 \end{cases}, \tag{2}$$

where $I$ indicates the identity matrix. We then have that the hypothesis $h^*(X) = 1$ iff $x_1 \in (-3, 3)$ achieves the perfect classification on all domains. Let $\mathcal{D}_r, \mathcal{D}_b$ denote source domains and $\mathcal{D}_o, \mathcal{D}_g$ denote target domains. Given a feature transformation $g(X) = \mathbb{I}_{x_1 < 0} \cdot (x_1 + 3) + \mathbb{I}_{x_1 > 0} \cdot (x_1 - 3)$ with the feature distribution $\mathcal{D}_{rb} = g \circ \mathcal{D}_b = g \circ \mathcal{D}_r = \mathcal{N}([0, -3], I)$. Namely, the invariant representation is learnt, which is $\mathcal{D}_{rb}$. However, the labeling functions $f_r$ of $\mathcal{D}_r$ and $f_b$ of $\mathcal{D}_b$ are just the reverse such that $f_r(X) = 1 - f_b(X); \forall X \in \mathcal{D}_{rb}$. In this case, we then have $\epsilon_{rb}(h) = 1$ (see details of derivation in Appendix A.1). In other words, for any hypothesis $h$, the invariant representation leads to large joint errors even on all source domains, let alone unseen target domains. Therefore, given this counterexample, DG methods without considering labeling function shift are fragmentary, however, as far as we know, no study on DG focus on such a challenging problem. To make clear how labeling function shifts influence generalization performance, we prove a novel domain generalization error bound by explicitly considering the labeling function shift in the next subsection.

## 2.2 Domain Generalization Bound

We first provide an upper bound that directly adapted from Ben-David et al. (2006); Zhang et al. (2021a).

**Proposition 1** *Let $\mathcal{H}$ be a hypothesis space and denote $\tilde{\mathcal{D}}$ as the induced distribution over feature space $\mathcal{Z}$ for every distribution $\mathcal{D}$ over raw space. Define $\mathcal{D}_i$ as a source distribution over $\mathcal{X}$, which enables a mixture construction of source domains as $\mathcal{D}_\alpha = \sum_{i=1}^K \alpha_i \mathcal{D}_i(\cdot)$. Denote a fictitious distribution $\mathcal{D}_\mathcal{T}^\alpha = \sum_{i=1}^K \alpha_i^* \mathcal{D}_i(\cdot)$ as the convex combination of source domains which is the closest to $\mathcal{D}_\mathcal{T}$, where $\alpha_1^*, ..., \alpha_K^* = \arg\min_{\alpha_1,...,\alpha_K} d_\mathcal{H}(\mathcal{D}_\mathcal{T}, \sum_{i=1}^K \alpha_i \mathcal{D}_i(\cdot))$. The fictitious distribution induces a feature space distribution $\tilde{\mathcal{D}}_\mathcal{T}^\alpha = \sum_{i=1}^K \alpha_i^* \tilde{\mathcal{D}}_i(\cdot)$. The following inequality holds for the risk $\epsilon_\mathcal{T}(h)$ on any unseen target domain $\mathcal{D}_\mathcal{T}$ (see*

*appendix A.2 for detailed derivations and explanations):*

$$\epsilon_{\mathcal{T}}(h) \leq \lambda_\alpha + \sum_{i=1}^{K} \alpha_i \epsilon_i(h) + d_{\mathcal{H}}(\tilde{\mathcal{D}}_{\mathcal{T}}^\alpha, \tilde{\mathcal{D}}_\alpha) + d_{\mathcal{H}}(\tilde{\mathcal{D}}_{\mathcal{T}}, \tilde{\mathcal{D}}_{\mathcal{T}}^\alpha). \tag{3}$$

Many recent works on DG via learning invariant representations can get intuition from the above analysis Li et al. (2018b); Ganin et al. (2016); Li et al. (2018c). Specifically, a parametric feature transformation $g : \mathcal{X} \to \mathcal{Z}$ is learnt such that the induced source distributions on $\mathcal{Z}$ are close to each other. $g$ is called an *invariant representation w.r.t* $\mathcal{H}$ if $d_{\mathcal{H}}(\tilde{\mathcal{D}}_i, \tilde{\mathcal{D}}_j) = 0, \forall i, j \in [1, 2, ..., K]$. Besides, a hypothesis $h$ over the feature space $\mathcal{Z}$ is found to achieve small empirical errors on source domains. However, the bound also depends on the risk of the optimal hypothesis $\lambda_\alpha$, which is usually intractable to compute for most practical hypothesis spaces and makes the bound conservative and loose in many cases. In this subsection, inspired by the counterexample, we provide a tighter upper bound for DG that considers labeling function shifts as follows.

**Proposition 2** *Let $\{\mathcal{D}_i, f_i\}_{i=1}^{K}$ and $\mathcal{D}_{\mathcal{T}}, f_{\mathcal{T}}$ be the empirical distributions and corresponding labeling function. For any hypothesis $h \in \mathcal{H}$ and transformation $g$, given mixed weights $\{\alpha_i\}_{i=1}^{K}; \sum_{i=1}^{K} \alpha_i = 1, \alpha_i \geq 0$, we have:*

$$\epsilon_{\mathcal{T}}(h \circ g) \leq \sum_{i=1}^{K} \left( \mathbb{E}_{X \sim \mathcal{D}_i} \left[ \alpha_i \frac{P_{\mathcal{T}}(X)}{P_i(X)} |h \circ g - f_i| \right] + \alpha_i \mathbb{E}_{\mathcal{D}_{\mathcal{T}}} [|f_i - f_{\mathcal{T}}|] \right). \tag{4}$$

*Proof: See Appendix A.3.*

The above two terms in the upper bound have natural interpretations: the first term is the weighted source errors, the second one measures the distance between the labeling functions from the source domain and target domain. Compared to Eq. equation 3, Eq. equation 4 does not depend on $\lambda_\alpha$, namely, the choice of the hypothesis class $\mathcal{H}$ makes no difference. Besides, the new upper bound in Eq. equation 4 reflects the influence of labeling function shifts and is independent of invariant representations.

**Remark**. Eq. (4) provides a new intuition on the design of DG models. Specifically, labeling functions $f_i, f_{\mathcal{T}}$ and density ratios $\frac{P_{\mathcal{T}}(x)}{P_i(x)}$ are constant and cannot be optimized. Therefore, we focus on mixed weights $\alpha_i$ and $h \circ g$. The first term will be minimized when $h \circ g$ attains low errors in source domains. The second term cannot be optimized directly, however, we can manipulate $\alpha$ to affect this term as follows. given $f_{\mathcal{T}}$, if we can find the source domain $\mathcal{D}_{i^*}$ with a labeling function $f_{i^*}$ that minimizes $\mathbb{E}_{\mathcal{T}}[|f_{i^*} - f_{\mathcal{T}}|]$, then we have that $\alpha_i = 1$, iff $i = i^*$, otherwise 0 makes this term the minimum. As a whole algorithm, these two procedures correspond to simultaneously finding the domain $\mathcal{D}_{i^*}$ whose labeling function is close to $f_{\mathcal{T}}$, setting $\alpha_{i^*} = 1$ and learning $h \circ g$ on $\mathcal{D}_{i^*}$ to minimize the source error. Namely, as long as we can accurately estimate $\mathbb{E}_{\mathcal{T}}[|f_i - f_{\mathcal{T}}|]$, only one domain is required for training to minimize the error in the target domain. However, calculating $\mathbb{E}_{\mathcal{T}}[|f_{i^*} - f_{\mathcal{T}}|]$ is intractable especially when $\mathcal{D}_{\mathcal{T}}$ is unseen during training. To tackle the challenge and follow the intuition brought by Eq. equation 4, we propose a new Domain-Specific Risk Minimization (DRM) method for domain generalization.

## 3 Domain-Specific Risk Minimization

The main training and testing pipelines using the proposed Domain-Specific Risk Minimization (DRM) are shown in Figure 2. One of our main contributions is the modeling of **domain-specific labeling function**. Specifically, given $K$ source domains, DRM utilizes a shared encoder $g$ and a group of classifiers $\{h_i\}_{i=1}^{K}$ for all domains, respectively. The encoder is trained by all data samples while each classifier $h_i$ is trained by using only images from the domain $\mathcal{D}_i$. In this way, DRM can attain 0 source error in the above-mentioned counterexample by using $g(X) = X$ and

$$h_r(X) = \begin{cases} 0 & x_1 \leq -3 \\ 1 & x_1 > -3 \end{cases}, \quad h_b(X) = \begin{cases} 1 & x_1 \leq 3 \\ 0 & x_1 > 3 \end{cases}.$$

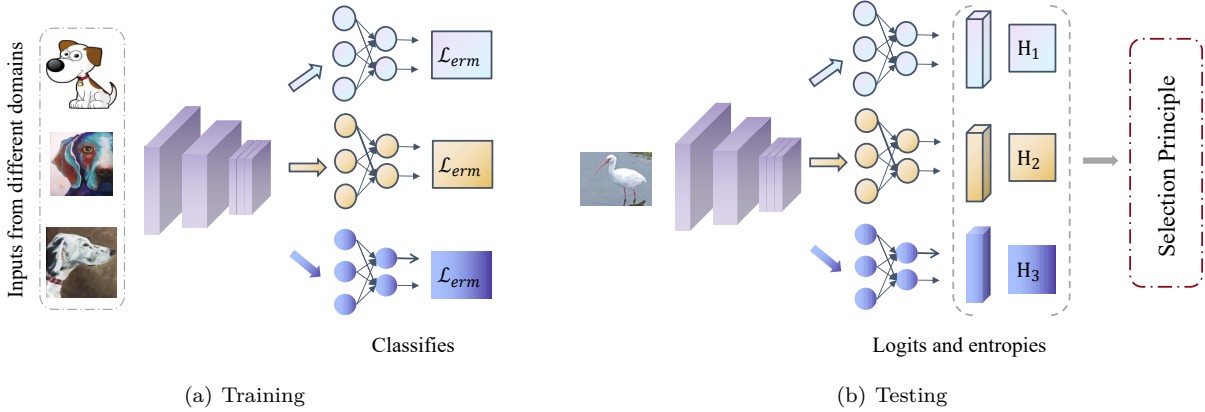

(a) Training                                           (b) Testing

Figure 2: An illustration of the training and testing pipelines using DRM. (a) during training, it jointly optimizes an encoder shared by all domains and the specific classifiers for each individual domain. $\mathcal{L}_{erm}$ indicates the cross-entropy loss function. (b) the new image is first classified by all classifiers and the entropy $H_i$ $(i = 1, \ldots, K)$ are then calculated. The logits with the minimum entropy will be used for the final prediction.

Furthermore, the choice of $g$ is not a matter and we can easily generalize it to other cases. For example, given $g(X) = \mathbb{I}_{x_1<0} \cdot (x_1 + 3) + \mathbb{I}_{x_1>0} \cdot (x_1 - 3)$ for invariant representation. DRM can still attain 0 source error by using

$$h_r(X) = \begin{cases} 0 & x_1 \leq 0 \\ 1 & x_1 > 0 \end{cases} , \qquad h_b(X) = \begin{cases} 1 & x_1 \leq 0 \\ 0 & x_1 > 0 \end{cases} .$$

If we go back to Eq. equation 4, with domain-specific classifiers, we then have the bound

$$\sum_{i=1}^{K} \alpha_i \left( \mathbb{E}_{x \sim \mathcal{D}_i} \left[ \frac{P_{\mathcal{T}}(x)}{P_i(x)} |h_i \circ g - f_i| \right] + \mathbb{E}_{\mathcal{D}_{\mathcal{T}}}[|f_i - f_{\mathcal{T}}|] \right). \tag{5}$$

Therefore, Eq. equation 5 shows that it is rather possible to achieve low errors on source domains by using the domain-specific classifiers than just one hypothesis $h$. It is also possible but not efficient to use specific $h_i \circ g_i$ for each domain. Besides, we also observe that, on the Colored MNIST dataset, it achieves the generalization accuracy 64.8% when using specific $h_i \circ g_i$, while it is 70.1% for using specific $h_i$. A possible reason is that a shared encoder $g$ can be seen as an implicit regularization, which prevents the model from overfitting specific domains.

We do not aim at a lower source error but also want to know *"how to determine mixed weights $\alpha$ such that low target domain error can be achieved?"*. To answer this question, we devise a **minimum entropy selection strategy** as follows. Given the following two assumptions: *"the learnt $h_i \circ g$ can well approximate $f_i$"* and *"the more confident prediction $h_i \circ g$ makes on $\mathcal{D}_{\mathcal{T}}$, the more similar $f_i$ and $f_{\mathcal{T}}$ will be"*. We then have, during testing, the $K$ individual classification logits as $\{\bar{\mathbf{y}}^k\}_{k=1}^{K}$, where $\bar{\mathbf{y}}^k = [y_1^k, ..., y_c^k]$, and $c$ is the number of classes. Then, the prediction entropy of $\bar{\mathbf{y}}^k$ can be calculated as

$$H_k = -\sum_{i=1}^{c} \frac{y_i^k}{\sum_{j=1}^{c} y_j^k} \log \frac{y_i^k}{\sum_{j=1}^{c} y_j^k}. \tag{6}$$

We choose the logit with the minimum entropy as the final prediction, namely $i^* = \arg\min\{H_i\}_{i=1}^{K}$ and $\alpha_{i^*=1}$. In our experiments, we find that the prediction entropy is consistent with the domain similarity, *e.g.*, in the counterexample, $\mathcal{D}_o$ is more similar to $\mathcal{D}_r$ than to $\mathcal{D}_b$, hence the entropy when $X \in \mathcal{D}_o$ is classified by $h_r$ is less than the entropy classified by $h_b$. In this way, Figure 1(c) shows that the learnt classification boundaries can attain 0 test errors on both the unseen target domains $\mathcal{D}_o$ and $\mathcal{D}_g$. Note that not all datasets have a significant visual difference between domains, and the prediction entropy is not exactly equivalent

to the labeling function difference. A one-hot mixed weight is too deterministic and cannot fully utilize all learned classifiers. **Softing mixed weights** can further boost generalization performance, *i.e.,* we generate the final prediction as

$$\sum_{k=1}^{K} \bar{\mathbf{y}}_k \frac{H_k^{-\gamma}}{\sum_{i=1}^{K} H_i^{-\gamma}}, \tag{7}$$

where $H_k^{-\gamma}$ indicates the contribution of each classifier. Specifically, for $\gamma = 0$, we then have a uniform combination, *i.e.,* $\alpha_i = 1/K, \forall i \in [1, 2, ..., K]$; for $\gamma \to \infty$, we then have a one-hot weight vector with $\alpha_i = 1$ iff $i = i^*$ otherwise 0.

**Remark.** By modeling domain-specific labeling functions, DRM can further reduce source errors (*i.e.,* the first term in our upper bound); For the second term, the entropy-based selection strategy allows us to select appropriate mixed weights and avoid directly calculating labeling function difference.

### 3.1 Case Studies

In this subsection, we perform case study analysis on the Colored MNIST dataset Arjovsky et al. (2019), where spurious correlations are manually created and can thus be a good indicator, to verify the following remarks:

- *DRM has better generalizability than invariant learning-based methods.*

- *DRM retains high accuracies on source domains and is orthogonal to invariant learning-based methods.*

- *DRM implicitly reduces prediction entropy and the entropy-based strategy performs well on finding a proper labeling function for inference.*

As shown in Table 1, ERM achieves high accuracies on training domains but below-chance accuracy on the test domain due to relying on the spurious correlations. IRM forms a tradeoff between training and testing accuracy Arjovsky et al. (2019). An ERM model trained on only gray images, *i.e.,* ERM (gray), is perfectly invariant by construction, and attains a better tradeoff than IRM. The upper bound performance of invariant representations (OIM) is a hypothetical model that not only knows all spurious correlations but also has no modeling capability limit. For averaged generalization performance, DRM, without any invariance regularization, outperforms IRM by a large margin (more than 2.4%). Besides, the training accuracy attained by DRM is even higher than ERM and significantly higher than IRM and OIM. Note that DRM is complementary with invariant learning-based methods, where incorporating CORAL Sun & Saenko (2016) can further boost both training and testing performances. Though the Colored MNIST dataset is a good indicator to show the model capacity for avoiding spurious correlation, these spurious correlations therein are unrealistic and utopian. Therefore, when testing on large DG benchmarks (*e.g.,* PACS, VLCS, DomainNet), ERM outperforms IRM. Different from them, DRM not only performs well on the semi-synthetic dataset but also attains state-of-the-art performance on large benchmarks.

The prediction entropy is often related to the fact that more confident predictions tend to be correct Wang et al. (2021). In Figure 3(a), we find that the entropy in target domain ($d = 2$) tends to be greater than the entropy in source domains, where the source domain with stronger spurious correlations ($d = 1$) also has larger entropy than easier one ($d = 0$). Fortunately, with the entropy minimization strategy, we can find the most confident classifier for a given data sample, and DRM can reduce the entropy of predictions (Figure 3(b)). To further analyze the entropy minimization strategy, we visualize the domain-classifier correlation matrix in Figure 3(c), where the entropy between the domain and its corresponding classifier is minimal, verifying the efficiency of the entropy minimization strategy. Please refer to Section 4.3 for more analysis on the domain-classifier correlation matrix.

## 4 Experiments

In this section, we evaluate the proposed DRM on several popular DG benchmarks. We also perform detailed ablation studies and analyses to better understand the proposed method.

| Method | +90% ($d=0$) train | test | +80% ($d=1$) train | test | -90% ($d=2$) train | test | Avg train | test |
|---|---|---|---|---|---|---|---|---|
| ERM | 86.1±3.9 | 71.8±0.4 | 83.6±0.5 | 72.9±0.1 | 87.5±3.4 | 28.7±0.5 | 85.7 | 57.8 |
| IRM | 78.2±9.5 | 72.0±0.1 | 70.6±9.1 | 72.5±0.3 | 85.3±4.7 | 58.5±3.3 | 78 | 67.7 |
| **DRM** | 81.8±9.8 | **86.7±2.4** | 90.2±0.2 | 80.6±0.2 | 88.0±4.5 | 43.1±7.5 | 86.7 | 70.1 |
| **DRM+CORAL** | 83.4±8.6 | 85.3±2.3 | **91.6±0.7** | **80.7±0.2** | **89.4±4.9** | 47.2±3.6 | **88.1** | **71.1** |
| RG | 50 | 50 | 50 | 50 | 50 | 50 | 50 | 50 |
| OIM | 75 | 75 | 75 | 75 | 75 | 75 | 75 | 75 |
| ERM (gray) | 84.8±2.7 | 73.9±0.3 | 84.3±1.4 | 73.7±0.4 | 83.4±2.3 | 73.8±0.7 | 84.2 | 73.8 |

Table 1: Accuracies (%) of different methods on training/testing domains for the Colored MNIST synthetic task. OIM (optimal invariant model) and RG (random guess) are hypothetical mechanisms.

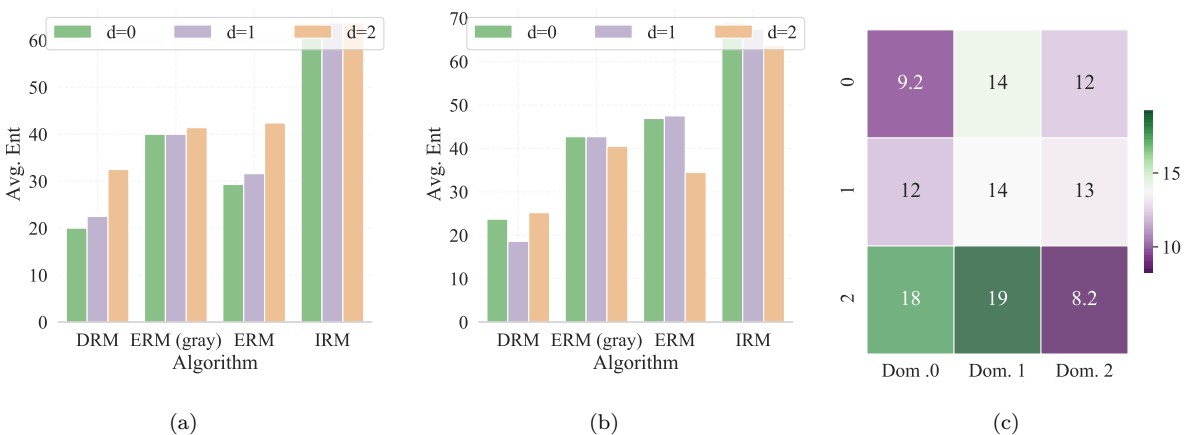

(a)  (b)  (c)

Figure 3: The entropy of different predictions. (a) Training domain $\{0,1\}$ and testing domain $\{2\}$. (b) The average of training/testing domains $\{0,1\}/\{2\}$, $\{0,2\}/\{1\}$, and $\{1,2\}/\{0\}$. (c) Domain-classifier correlation matrix, the value $v_{ij}$ is the entropy of predictions incurred by predicting samples in domain $i$ with classifier $j$. Dom.$i$ indicates the classifier for the domain $d=i$.

## 4.1 Experimental Setup

We use four popular domain generalization benchmark datasets: Colored MNIST Arjovsky et al. (2019), Rotated MNIST Ghifary et al. (2015), PACS Li et al. (2017), VLCS Torralba & Efros (2011), and Domain-Net Peng et al. (2019). We compare our model with ERM Vapnik (1999), IRM Arjovsky et al. (2019), Mixup Yan et al. (2020), MLDG Li et al. (2018a), CORAL Sun & Saenko (2016), DANN Ganin et al. (2016), CDANN Li et al. (2018c), MTL Blanchard et al. (2021), SagNet Nam et al. (2021), ARM Zhang et al. (2021b), VREx Krueger et al. (2021), RSC Huang et al., Fish Shi et al. (2022), and Fishr Rame et al. (2021). Following Gulrajani & Lopez-Paz (2021), we use the same backbone network, ConvNet for Rotated MNIST and Colored MNIST, and ResNet-50 for the remaining datasets. For fair comparison, we use similar hyperparameters to Gulrajani & Lopez-Paz (2021) for all other datasets except the DomainNet dataset, where we observe that the proposed DRM does not converge within $5k$ iterations and we thus train it with an extra of $5k$ iterations.

## 4.2 Results

In this subsection, we report the performance of DRM under different generalization settings as follows.

**Domain generalization.** The average domain generalization results on all benchmarks are shown in Table 2. We observe consistent improvements achieved by DRM compared to existing algorithms, where the

| Method | Colored MNIST | Rotated MNIST | VLCS | PACS | DomainNet | Avg |
|---|---|---|---|---|---|---|
| ERM Vapnik (1999) | $57.8 \pm 0.2$ | $97.8 \pm 0.1$ | $77.6 \pm 0.3$ | $86.7 \pm 0.3$ | $41.3 \pm 0.1$ | 72.2 |
| IRM Arjovsky et al. (2019) | $67.7 \pm 1.2$ | $97.5 \pm 0.2$ | $76.9 \pm 0.6$ | $84.5 \pm 1.1$ | $28.0 \pm 5.1$ | 70.9 |
| Mixup Yan et al. (2020) | $58.4 \pm 0.2$ | $98.0 \pm 0.1$ | $78.1 \pm 0.3$ | $86.8 \pm 0.3$ | $39.6 \pm 0.1$ | 72.2 |
| MLDG Li et al. (2018a) | $58.2 \pm 0.4$ | $97.8 \pm 0.1$ | $77.5 \pm 0.1$ | $86.8 \pm 0.4$ | $41.6 \pm 0.1$ | 72.4 |
| CORAL Sun & Saenko (2016) | $58.6 \pm 0.5$ | $98.0 \pm 0.0$ | $77.7 \pm 0.2$ | $87.1 \pm 0.5$ | $41.8 \pm 0.1$ | 72.6 |
| DANN Ganin et al. (2016) | $57.0 \pm 1.0$ | $97.9 \pm 0.1$ | $79.7 \pm 0.5$ | $85.2 \pm 0.2$ | $38.3 \pm 0.1$ | 71.6 |
| CDANN Li et al. (2018c) | $59.5 \pm 2.0$ | $97.9 \pm 0.0$ | $79.9 \pm 0.2$ | $85.8 \pm 0.8$ | $38.5 \pm 0.2$ | 72.3 |
| MTL Blanchard et al. (2021) | $57.6 \pm 0.3$ | $97.9 \pm 0.1$ | $77.7 \pm 0.5$ | $86.7 \pm 0.2$ | $40.8 \pm 0.1$ | 72.1 |
| SagNet Nam et al. (2021) | $58.2 \pm 0.3$ | $97.9 \pm 0.0$ | $77.6 \pm 0.1$ | $86.4 \pm 0.4$ | $40.8 \pm 0.2$ | 72.2 |
| ARM Zhang et al. (2021b) | $63.2 \pm 0.7$ | $98.1 \pm 0.1$ | $77.8 \pm 0.3$ | $85.8 \pm 0.2$ | $36.0 \pm 0.2$ | 72.2 |
| VREx Krueger et al. (2021) | $67.0 \pm 1.3$ | $97.9 \pm 0.1$ | $78.1 \pm 0.2$ | $87.2 \pm 0.6$ | $30.1 \pm 3.7$ | 72.1 |
| RSC Huang et al. | $58.5 \pm 0.5$ | $97.6 \pm 0.1$ | $77.8 \pm 0.6$ | $86.2 \pm 0.5$ | $38.9 \pm 0.6$ | 71.8 |
| Fish Shi et al. (2022) | $61.8 \pm 0.8$ | $97.9 \pm 0.1$ | $77.8 \pm 0.6$ | $85.8 \pm 0.6$ | $43.4 \pm 0.3$ | 73.3 |
| Fishr Rame et al. (2021) | $68.8 \pm 1.4$ | $97.8 \pm 0.1$ | $78.2 \pm 0.2$ | $86.9 \pm 0.2$ | $41.8 \pm 0.2$ | 74.7 |
| **DRM** | $70.1 \pm 2.0$ | $98.1 \pm 0.2$ | $\mathbf{80.0 \pm 0.3}$ | $\mathbf{87.5 \pm 1.2}$ | $42.4 \pm 0.1$ | 75.6 |
| **DRM+CORAL** | $\mathbf{71.1 \pm 1.7}$ | $\mathbf{98.3 \pm 0.1}$ | $79.0 \pm 2.4$ | $87.4 \pm 0.9$ | $\mathbf{42.7 \pm 0.1}$ | **75.7** |

Table 2: Out-of-distribution generalization performance.

| | **Rotated MNIST** | | | | | | |
|---|---|---|---|---|---|---|---|
| Method | 0 | 15 | 30 | 45 | 60 | 75 | Avg |
| ERM | **99.1±0.2** | 98.8±0.5 | 99.0±0.1 | **99.1±0.2** | 99.0±0.2 | 98.9±0.4 | 99.0 |
| IRM | 92.9±1.8 | 92.6±2.5 | 94.7±1.0 | 89.9±1.5 | 92.1±2.2 | 94.9±1.5 | 92.9 |
| **DRM(ours)** | 99.0±0.2 | **99.0±0.3** | **99.0±0.3** | 99.0±0.2 | **99.1±0.2** | **99.0±0.2** | **99.0** |
| | **DomainNet** | | | | | | |
| Method | clip | info | paint | quick | real | sketch | Avg |
| ERM | **50.4±11.4** | 58.3±6.2 | **53.4±12.6** | 54.6±12.7 | **50.8±11.0** | 51.9±12.6 | 53.2 |
| IRM | 33.4±4.1 | 53.2±1.4 | 34.0±4.1 | 35.1±3.4 | 33.0±3.8 | 31.5±3.1 | 36.7 |
| **DRM(ours)** | 50.1±14.3 | **58.3±10.4** | 52.5±14.7 | **58.1±10.3** | 50.2±13.2 | **52.1±11.5** | **53.6** |

Table 3: In-distribution performance on Rotated MNIST and DomainNet.

average accuracy achieved by DRM (75.6%) clearly outperforms all other methods. This result suggests that DRM has strong generalization capability on not only small semi-synthesis datasets but also large real-world benchmarks. Refer to Appendix C for generalization results on every domain of all benchmarks.

**Source domain accuracy.** Current DG methods do not consider keeping very good performance on source domains, while it is also of great importance in real-world applications Yang et al. (2021). Taking the performance on source domain into account, we then show the in-distribution performances of Rotated MNIST and DomainNet in Table 3, and VLCS and PACS in Table 4. Specifically, here we compare the classification performances (*e.g.,* on PACS, when 'P' is chosen as the unseen target domain, the average classification performance on other domains is reported). DRM achieves comparable or even superior performance on source domains compared to ERM and beats IRM by a large margin.

| Method | **VLCS** | | | | | **PACS** | | | | |
|---|---|---|---|---|---|---|---|---|---|---|
| | C | L | S | V | Avg | A | C | P | S | Avg |
| ERM | 78.2±3.3 | 87.8±9.0 | 86.3±10.2 | 83.3±11.6 | 83.9 | 96.7±0.3 | 96.4±1.5 | **95.3±1.2** | **96.3±0.1** | 96.2 |
| IRM | 76.9±2.9 | **88.2±8.9** | 85.3±9.8 | 77.3±1.0 | 81.9 | 95.9±1.6 | 94.2±2.5 | 94.3±1.0 | 94.5±1.8 | 94.7 |
| **DRM(ours)** | **78.5±2.9** | 87.2±9.2 | **87.3±9.0** | **84.0±10.9** | **84.3** | **96.9±0.3** | **96.4±1.3** | 95.2±0.9 | 96.1±0.6 | **96.2** |

Table 4: In-distribution performance on VLCS and PACS.

**Multi-domain generalization.** IRM Arjovsky et al. (2019) introduces specific conditions for an upper bound on the number of training environments required such that an invariant optimal model can be obtained, which stresses the importance of several training environments. In this paper, we reduce the training

| | Rotated MNIST | | | | | | |
| | Target domains $\{0, 30, 60\}$ | | | Target domains $\{15, 45, 75\}$ | | | |
| Method | 0 | 30 | 60 | 15 | 45 | 75 | Avg |
|---|---|---|---|---|---|---|---|
| ERM | 96.0±0.3 | 98.8±0.4 | 98.7±0.1 | 98.8±0.3 | **99.1±0.1** | 96.7±0.3 | 98.0 |
| IRM | 80.9±3.2 | 94.7±0.9 | 94.3±1.3 | 94.3±0.8 | 95.5±0.5 | 91.1±3.1 | 91.8 |
| **DRM(ours)** | **97.1±0.2** | **98.8±0.2** | **98.9±0.3** | **98.8±0.1** | 98.8±0.0 | **98.1±0.7** | **98.4** |

Table 5: Generalization performance on multiple unseen target domains.

environments on the Rotated MNIST from five to three. As shown in Table 5, as the number of training environment decreases, the performance of IRM fall sharply (*e.g.,* the averaged accuracy from 97.5% to 91.8%), and the performance on the most challenging domains $d = \{0, 5\}$ decline the most (94.9% → 80.9% and 95.2% → 91.1%). In contrast, both ERM and DRM retain high generalization performances while DRM outperforms ERM on domains $d = \{0, 5\}$.

### 4.3 Ablation Studies and Analysis

**Correlation matrix.** From the correlation matrices, we find that: (i) The entropy of predictions between one source domain and its corresponding classifier is minimal. (ii) On the target domain, classifiers cannot attain very low entropy as they attained on the corresponding source domains. (iii) The entropy of predictions has a certain correlation with domain similarity. For example, in Figure 5(a), classifier for domain $d = 1$ (with rotation angle 15°) attains the minimum entropy on the unseen target domain $d = 0$ (no rotation). As the rotation angle increases, the entropy also increases. This phenomena has also occurred in other domains. Please refer to Figure 9

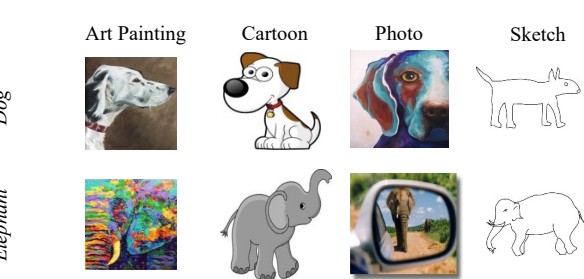

Figure 4: Some examples from PACS.

for more correlation matrices. We also visualize some examples from the PACS dataset in Figure 4, where we can see that the style of $d = 3$ is more similar to $d = 1$, and $d = 2$ is very dissimilar to $d = \{1, 3\}$, *etc.* Almost all of these similarity characters can be seen from the correlation matrix in Figure 5(b), namely, domain similarity encoded in the correlation matrices is consistent with our visual common sense.

**Softing mixed weights.** Figure 7 shows ablation experiments of hyper-parameter $\gamma$ on three benchmarks. Different benchmarks show different preferences on $\gamma$. For easy benchmarks Rotated MNIST and Colored MNIST, softing mixed weights is needless. The reason behinds this phenomenon can be found in Figure 5(a), the optimal classifier for target domain 0 of Rotated MNIST is exactly the classifier 1 and the prediction entropies will increase as the rotation angle increases. Hence, selecting the most approximate classifier based on the minimum entropy selection strategy is enough to attain superior generalization results. However, prediction entropies on other larger benchmarks, *e.g.,* VLCS, are not so regular as on the Rotated MNIST. On realistic benchmarks, a mixing of classifiers can bring some improvements. Besides, normalization, which is a method to reduce classification confidence[2], is also needless for semi-synthetic datasets (Rotated MNIST and Colored MNIST) and valuable for realistic benchmarks.

**Model complexity.** As shown in Table 6, methods that require manipulating gradients (Fish Shi et al. (2022)) or following the meta-learning pipeline (ARM Zhang et al. (2021b)) have much slower training speed compared to ERM. The proposed DRM, without the need for aligning representations Ganin et al. (2016); Li et al. (2018c), matching gradient Shi et al. (2022), or learning invariant representations Arjovsky et al. (2019); Zhang et al. (2021b), has a training speed that is faster than most existing DG

---

[2]Given two classification results from 2 classifiers $[2.1, 0.4, 0.5], [0.3, 0.6, 0.1]$ and assume the weights are all 1. The result is $[2.4, 1.0, 0.6]$ with normalization and $[1.0, 0.73, 0.27]$ without normalization. The former is more confident than the latter.

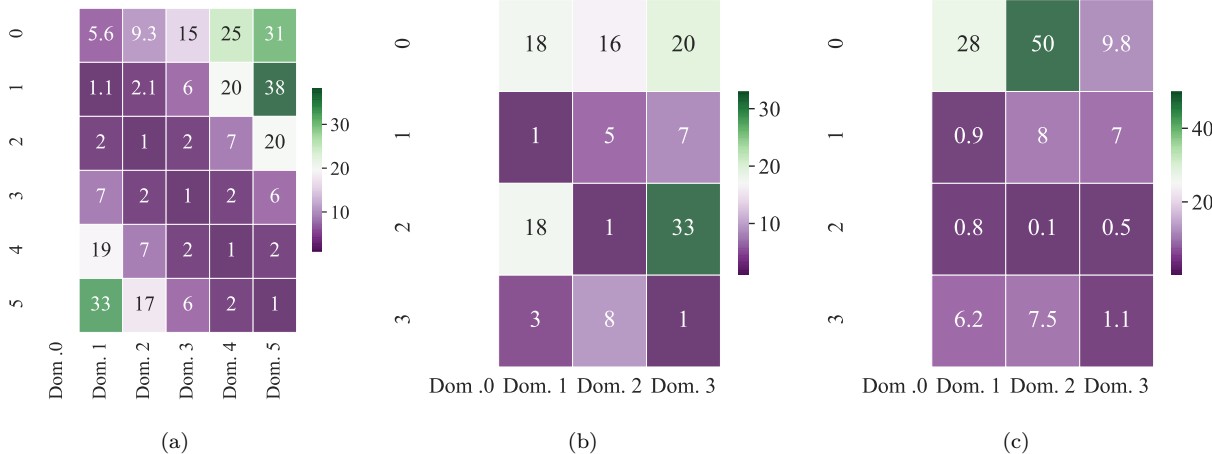

(a)            (b)            (c)

Figure 5: Domain-classifier correlation matrices on (a) Rotated MNIST, (b) PACS, and (c) VLCS datasets. Domain $d = 0$ as the target domain.

| Method | Colored MNIST | | Rotated MNIST | | PACS | |
|---|---|---|---|---|---|---|
| | Time (sec) | # Params | Time (sec) | # Params | Time (sec) | # Params |
| ERM | 71.02 | 0.3542M | 168.32 | 0.3546M | 2,717.5 | 22.4326M |
| IRM | 101.49 | 0.3542M | 236.8 | 0.3546M | 2,786.3 | 22.4326M |
| CDANN | 89.14 | 0.4492M | 191.61 | 0.4513M | 2,744.8 | 23.01M |
| ARM | 161.51 | 0.4573M | 360.69 | 0.4562M | 6,616.9 | 22.5398M |
| FISH | 137.17 | 0.3542M | 251.76 | 0.3546M | 23,849.5 | 22.4326M |
| **DRM(ours)** | 83.39 | 0.3544M | 203.15 | 0.3595M | 2,895.1 | 22.46M |

Table 6: Comparisons of different methods on number of parameters and training time.

methods especially on small datasets (ColoredMNIST and RotatedMNIST). The training speed of DRM is slower than ERM because of the need for training additional $K-1$ classifiers. As the number of domains/classes increases or the feature dimension increases, the training time of DRM will increase accordingly, however, DRM is always comparable to ERM and much faster than Fish and ARM. For model parameters, since all classifiers in our implementation are just a linear layer, the total parameters of DRM is similar to ERM and much less than existing methods such as CDANN and ARM.

**Convergence analysis.** The training dynamics of DRM and several baselines on PACS dataset are shown in Figure 6, where $d = 0$ is the target domain. We can see that, due to the adversarial training nature, Domain adversarial training method (DANN) is highly unstable and hard to converge. IRM has a similar pattern yet is more stable. ARM follows a meta-learning pipeline and converges slowly. In contrast, DRM converges even faster than ERM thanks to the specific classifiers.

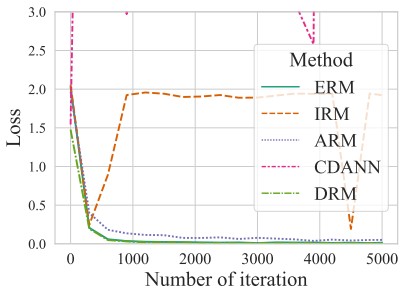

Figure 6: Loss curves.

## 5 Related work

**Domain adaptation and domain generalization** Domain/Out-of-distribution generalization (Muandet et al., 2013; Sagawa et al., 2019; Li et al., 2018a; Blanchard et al., 2021; Li et al., 2018c; Zhang et al., 2021a;

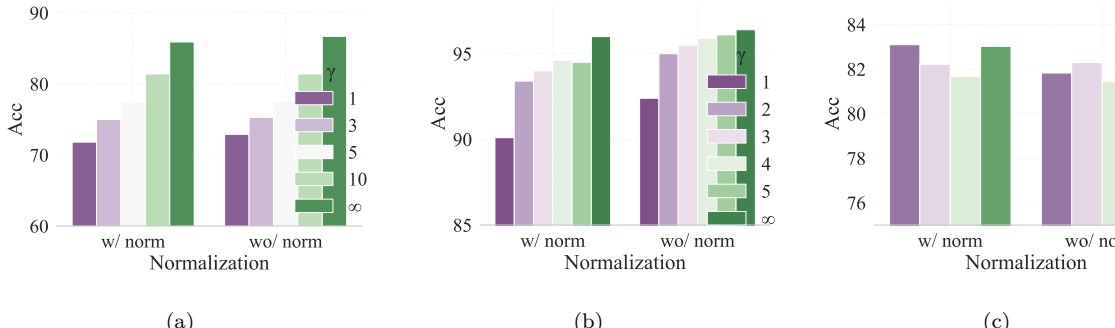

(a)             (b)             (c)

Figure 7: Different mixing weights on the (a) Colored MNIST (target domain $d = 2$) (b) Rotated MNIST (target domain $d = 0$), and (c) PACS datasets (target domain $d = 3$). Given a classification vector $\bar{\mathbf{y}} = [y_1, y_2, ..., y_c]$, $c$ is the number of classes, performing normalization means that let $y_i = y_i / \sum_{j=1}^{c} y_j$ before mixing.

2022) aims to learn a model that can extrapolate well in unseen environments. Representative methods like Invariant Risk Minimization (IRM) (Arjovsky et al., 2019) and its variant (Ahuja et al., 2020) are recently proposed to tackle this challenge. IRM centers on the objective of extracting data representations that lead to invariant prediction across environments under a multi-environment setting. In this paper, we emphasize the importance of labeling function modeling and show that, even without an invariance strategy, the proposed DRM can attain superior generalization capacity.

**Labeling function shift and multi classifiers**. Labeling function shift or conditional shift is not a novel concept and is commonly used in domain adaptation Zhao et al. (2019); Stojanov et al. (2021); Zhang et al. (2013). There are also some studies on DG that take into account this problem. CDANNLi et al. (2018c) considers the scenario where both $P(X)$ and $P(Y|X)$ change across domains and propose to learn a conditional invariant neural network to minimize the discrepancy in $P(X|Y)$ across different domains. Liu et al. (2021) explores both the conditional and label shifts in DG and aligns the conditional shift via the variational Bayesian inference. The proposed DRM is different from these studies because we want the labeling functions $P(Y|X)$ more specific to each domain rather than invariant.

**Ensemble learning in domain generalization** learn ensembles of multiple specific models for different source domains to improve the generalization ability, *e.g.,* domain-specific neural networks layer Ding & Fu (2017), domain-specific classifiers Wang et al. (2020), and domain-specific batch normalization Segu et al. (2020). Domain-specific classifiers are also used in this work, however, classifiers ensemble is not important for DRM. In contrast, the proposed classifier selection strategy can select the most appropriate labeling function for prediction and attains superior performance to baselines.

## 6 Conclusion

In this paper, we study the important problem of labeling function shifts for domain generalization theoretically and empirically. We first construct an example to show that learning an invariant representation without considering the labeling function shift is not sufficient for a good generalization. We then prove a novel upper bound for the target error, which motivates us to propose DRM to eliminate the negative effects brought by labeling function shifts. DRM achieves not only a superior generalization performance but also maintain low source errors simultaneously. We hope that our results can shed new light on the model design for domain generalization problems. One possible direction is to estimate $\alpha_i \frac{P_T(x)}{P_i(x)}$ and then reweight data samples, which will be the subject of our future study. In addition, the minimum entropy selection strategy is used intuitively but lacks a theoretical connection to the labeling function difference. Hence another possible direction is to improve current strategy or propose a new strategy that is not only theoretically rigorous but also empirically works better.

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
