# OpenReview forum: "Exploring Conditional Shifts for Domain Generalization"
_TMLR — Withdrawn by Authors_

### Review · Reviewer_dAoQ · 2022-06-24

**Summary Of Contributions:**

This paper studies the formalization of generalization error bound of domain generalization, the paper intends to introduce a new perspective that can lead to a new generalization error bound, and thus continue to lead to a new method.

**Broader Impact Concerns:**

None noted.

**Requested Changes:**

To summarize the discussions above (but please also refer to the above discussions for details)

- On the theoretical end
    - the discussion needs to clarify how it differentiates from the previous ones, hopefully in addition to extending the discussion from a DA setting to a DG setting.

- On the empirical end
   - please test the method with a more stringent and comprehensive benchmark.

**Strengths And Weaknesses:**

The paper is an interesting read and focuses on a very important problem that is in needs to be resolved. I think the paper has great potential after being further polished.

- Strengths

    - the paper introduces a new, interesting empirical method, and the method seemingly performs well on the empirical end.
    - the method part is written in a very straightforward manner, the empirical method is easy to understand
- Weakness
    - the theoretical discussion largely builds upon the counterexample, which has been discussed by (Zhao et al 2019), the main contribution of this paper is seemingly about extending the bound inspired from this counterexample from DA setting to DG setting. From this perspective, the theoretical contribution is quite limited.
    - the counterexample is essentially about the misaligned labeling function from source and target, other than (Zhao et al 2019), this example has also been widely discussed.
         - for example, the bound regarding labeling function has been discussed even in the earliest relevant literature (Theorem 1 in [1])
         - most recently, the community has explored more concrete formulations of what the misalignment of labeling function exactly is about [2], instead of the generic form.
         - although this paper is essentially about DG instead of DA, ignoring these discussions seemingly offers a blurred picture of the main theoretical contribution of this work.
    - While it's interesting to see an empirical method inspired by the theoretical discussion, one of the most important steps does not have any theoretical backup, but simply a heuristic choice: selecting the function based on entropy.
         - it could be helpful to demonstrate more ablation studies on how this choice matters on the empirical end.
         - it's quite unexpected that this method would work well across these settings, for example, in PACS, S is very different from the rest and all h_i will be working with data sufficiently different from the trained ones during test when S is the test domain. Why the method can still perform well needs to be discussed.
   - The empirical scope might be good a while ago, but I believe the DG community is rapidly exploring new testbed, I would recommend the authors to test with a more stringent testbed such as OOD-bench [3].

[1] Ben-David, S., Blitzer, J., Crammer, K., Kulesza, A., Pereira, F., & Vaughan, J. W. (2010). A theory of learning from different domains. Machine learning, 79(1), 151-175.

[2] Wang, H., Huang, Z., Zhang, H., & Xing, E. (2021). Toward learning human-aligned cross-domain robust models by countering misaligned features. arXiv preprint arXiv:2111.03740.

[3] Ye, Nanyang, et al. "OoD-Bench: Quantifying and Understanding Two Dimensions of Out-of-Distribution Generalization." Proceedings of the IEEE/CVF Conference on Computer Vision and Pattern Recognition. 2022.

---

### Review · Reviewer_j9CQ · 2022-06-30

**Summary Of Contributions:**

This paper explored the issues of learning invariant representation in domain generalization when the inherent label distributions are different across different sources. Then this paper derived a novel theoretical bound for explicitly illustrating the target risk, which inspired a novel approach in combining the source predictors. Further a novel selection rule DRM is proposed to select the relevant sources. Empirical validations on different benchmarks demonstrate its effectiveness.


**Broader Impact Concerns:**

N/A.

**Requested Changes:**

In general I like the idea. Based on the weak points,  I would suggest the following major revision:

- Additional discussion about counterexamples and theoretical results.
- Intuition and rationale of different assumptions within the paper.
- Additional experimental discussions and comparison on the ensemble-based approach.



**Strengths And Weaknesses:**

### Pros:
- This paper proposed a clear understanding of limitations in learning (marginal) invariant representation. Although several previous works implicitly discussed this issue in different contexts such as domain adaptation, this paper highlights the concerns in domain generalization.

- I would appreciate the authors for clearly illustrating the theoretical assumptions and limitations (e.g, Sec 3, Sec.6) of the proposed method. The proposed method is quite simple but effective.

- Extensive empirical results clearly validated the benefits of the proposed approach.

### Cons:
- There are several concerns in the counterexamples and theoretical results. In addition, some arguments are seemingly overclaimed.
- Concerns in proposed approach and certain experimental results.

### Detailed Comments

1. [About counterexamples]
- Indeed, the limitation of learning invariant representation has been highlighted in other contexts such as domain adaptation (such as Zhao 2019). The counterexample essentially does not provide additional information in understanding domain generalization. Besides, in the preliminary section, learning invariant representation is **not defined**, although it is defined later (below Equation (3)). I would suggest a clear definition in learning invariant representation.

- Another concern is the depth of the analysis. This paper does not provide sufficient discussion about learning invariant representation itself. E.g, for the representation function $g$ if have invariant P(g(X)) (in this paper), P(g(X)|Y) or P(Y|g(X)) across the sources, this also does not necessarily guarantee a small test risk. At this point, I really like the idea within this paper for a model selection during the test time for invariant P(g(X)). What about learning invariant P(g(X)|Y) or P(Y|g(X))? Is there a similar model selection approach? To this end, I would recommend a recent paper [1], which discussed the limitations in learning different kinds of invariance.


2. [About theoretical results]

- In Proposition 1. “However, the bound also depends on the risk of the optimal hypothesis, which is usually intractable to compute for most practical hypothesis spaces and makes the bound conservative and loose in many cases.”  This is partially incorrect since the joint optimal risk could be partially estimated (we have many sources with only one unknown target). If the source number is large, we enforce a joint minimum over the sources that could effectively control this term. (This idea will be analogue to IRM).  In addition, this theory should hold for many statistical divergences such as discrepancy. A discussion is expected.

- In proposition 2, it seems that there is no need to introduce the representation function $g$, we could simply prove a general predictors $h_i$ and derive an ensemble-based selection approach. Why not this idea? Note there is a small typo in the bound, it would be expressed as
$$\sum_{i=1}^{K} \alpha_i E_{D_1} $$

- Besides, in proposition 2, the density ratio is not properly discussed. I agree this is a constant, while it will influence the conventional ERM (it should be a density ratio weighted ERM). Simultaneously introducing the density ratio will induce an arbitrary large value where there is no common support of $P(x)$, i.e if x is in supp(P(x)) but out of supp(P_i(x)) with P_i(x)=0, then this bound could be fundamentally problematic. Note this is quite common in real images such as DomainNet. There exists an inconsistency between theory and practice. To this end, the bound should avoid introducing the density ratio, at least for P(x).

3. [About proposed approach]
- The minimum entropy selection strategy indeed relies on two assumptions. (1) the learnt predictor can well approximate” and (2) “the more confident prediction makes on T , the more similar fi and fT will be.” I have no concern on the first assumption with sufficient data in each source. While the second assumption made it quite difficult to follow. Is it possible to provide a formal description of assumption (2)? What is the rationale behind this assumption? In which scenarios does this assumption hold?

- From my understanding, intuitively, different source classifiers are used to predict the uncertainty of the target. If certain sources have confident predictions on the target, we should select such confident predictors. While this could be fundamentally incorrect. If the decision boundary is inverse, the high negative confidence will provide the worst predictor (This is exactly the assumption 2 states.) While I could not understand when assumption 2 holds. At least for the counterexample, it is still an impossible case….

4. [Experimental details] Tab 2, DRM+Coral, DRM in d=2, Test. It seems DRM still could not mitigate the spurious correlation. I could understand this may be because of the violation of assumption 2, the predictor has high confidence, while the ground truth labelling is still quite different. From this viewpoint, the proposed approach could not mitigate spurious correlations. Please note this does not indicate it could not improve the generalization property. In several real-world scenarios such as DomainNet, the decision boundary is similar but not inverse. This idea still works.  In Tab 6, the time and memory complexity could be better formatted.

5. [About ensemble ideas] I would suggest a detailed discussion (e,g, empirical comparison) about the proposed approach and ensemble approach. Since they shared several common points in theory and practice.

6. Minor: I would suggest merging the main paper and appendix for facilitating the reading.

Reference

[1] On the benefits of representation regularization in invariance based domain generalization. Machine Learning Journal, 2022.

---

### Review · Reviewer_XH5t · 2022-07-11

**Summary Of Contributions:**

This paper considers exploring the conditional shifts for DG. In the training phase, they consider many heads in the networks whose corresponding loss functions are different. In the testing phase, a selection procedure is proposed to select the good head to predict the labels of target domain data. Experiments indeed verify the performance of the proposed methods. Given the same-size model, they can achieve better performance.

**Broader Impact Concerns:**

No concerns on the ethical implications of the work.

**Requested Changes:**

See the weaknesses.

**Strengths And Weaknesses:**

Pros:

1. Experiments are solid. We can see the improvements on many datasets. The ablation study also verifies the effectiveness of the proposed model.

2. A theoretical insight is provided to show that conditional shift matters.

Cons:

1. On a high level, the key to why this method will work is unclear. Note that, if we add an additional layer to (a) in Figure 2, we can still consider that there is only one classifier (although it sets a rule to select a classifier when prediction). Thus, I did not see some novel parts that this method provides. The selection principle is just a classifier, nothing more. I hope for more explanations regarding this method.

2. Following 1, I am not sure if this paper overclaims its contribution. I think this paper might provide a better regularzier for DG, which can help learn a good general representation. If you do not learn a good representation, the selection procedure will not work well. TMLR is not a place emphasizing novelty, but it emphasizes the truth. Please consider the contribution clearly.

3. In the theory of DG, we care about when the problem can be solved. So, under which assumption, the DG problem can be solved (we can learn the f_T in the end) in your theoretical framework.

4. Some necessary conditions are missing in the theory of this paper. The ratio of densities (Prop. 2) is only well-defined when the densities are absolutely continuous. Be careful when you use the ratio of densities.

In general, this paper can be accepted after revision.

---

### Review · Reviewer_Gt5D · 2022-07-11

**Summary Of Contributions:**

The paper proposes a novel method for addressing the domain generalisation problem where there is a change in the labelling function between domains. Theoretical analysis showing how this change in the labelling functions impacts the achievable performance is provided. The authors benchmark their method on several common DG datasets, where it performs the best.

**Broader Impact Concerns:**

No concerns.

**Requested Changes:**

To justify including the theoretical analysis in the submission, it should be more tightly coupled with the proposed method.

**Strengths And Weaknesses:**

Strengths:

* The performance of the proposed method seems to be quite good, and comparisons are made to a extensive set of baselines. This, along with the correlation analysis (which I may not have interpreted correctly) provides some assurance that the method works well.

Weaknesses:

* While the proofs for the theorems seem to be correct, I have some significant concerns about the discussion surrounding the theoretical analysis:
  * Proposition 2 does actually depend on a similar quantity to $\\lambda_\\alpha$, because a poor choice of hypothesis class will have an impact on how well $|h \\circ g - f_i|$ can be minimised.
  * It seems like in at least some realistic cases, the density ratio will be undefined due to division by zero. E.g., if the source domain is sketches and the target domain is photos.
  * None of the terms in Proposition 2 are observable, because they are expected risks not empirical risks. This makes it difficult for me to see how one can obtain interesting conclusions from this line of analysis.
* The authors claim that the derivation of the proposed method follows from the intuition provided by the theoretical analysis, but this is not correct. In Section 3 the method is re-motivated on the basis of two assumptions: (i) realizability; and (ii) the positive correlation between classifier confidence and domain similarity.
* The clarity of the submission could be improved. For example:
  * How are the plots in Figure 5 generated? What does it mean to measure correlation between a "domain" and a "classifier"?
  * The authors say they use "similar" hyperparameters to DomainBed, but this implies they are different in some way. How are they different?

---

### Note · Authors · 2022-07-26

**Comment:**

I have read and agree with the venue's withdrawal policy on behalf of myself and my co-authors.

**Withdrawal Confirmation:**

I have read and agree with the venue's withdrawal policy on behalf of myself and my co-authors.